# Red Blood Cell Metabolism in Patients with Propionic Acidemia

Micaela Kalani Roy [1], Francesca Isabelle Cendali [1], Gabrielle Ooyama [2,3], Fabia Gamboni [1], Holmes Morton [2,3,*] and Angelo D'Alessandro [1,*]

1   Department of Biochemistry and Molecular Genetics, Anschutz Medical Campus, University of Colorado Denver, Aurora, CO 80045, USA; micaela.roy@cuanschutz.edu (M.K.R.); francesca.cendali@cuanschutz.edu (F.I.C.); fabia.gamboni@cuanschutz.edu (F.G.)
2   Central Pennsylvania Clinic, A Medical Home for Special Children & Adults, Belleville, PA 17004, USA; gabrielleooyama@gmail.com
3   Clinic for Special Children, Strasburg, PA 17579, USA
*   Correspondence: dholmesmorton@gmail.com (H.M.); angelo.dalessandro@cuanschutz.edu (A.D.); Tel.: +1-303-724-0096 (A.D.)

**Abstract:** Propionic acidemia (PA) is a rare autosomal recessive disorder with an estimated incidence of 1:100,000 live births in the general population. Due in part to an insufficient understanding of the disease's pathophysiology, PA is often associated with complications, and in severe cases can cause coma and death. Despite its association with hematologic disorders, PA's effect on red blood cell metabolism has not been described. Mass spectrometry-based metabolomics analyses were performed on RBCs from healthy controls ($n = 10$) and PKD patients ($n = 3$). PA was associated with a significant decrease in the steady state level of glycolytic products and the apparent activation of the PPP. The PA samples showed decreases in succinate and increases in the downstream dicarboxylates of the TCA cycle. BCAAs were lowered in the PA samples and C3 carnitine, a direct metabolite of propionic acid, was increased. Trends in the markers of oxidative stress including hypoxanthine, allantoate and spermidine were the opposite of those associated with elevated ROS burden. The alteration of short chain fatty acids, the accumulation of some medium chain and long chain fatty acids, and decreased markers of lipid peroxidation in the PA samples contrasted with previous research. Despite limitations from a small cohort, this study provides the first investigation of RBC metabolism in PA, paving the way for targeted investigations of the critical pathways found to be dysregulated in the context of this disease.

**Keywords:** erythrocyte; metabolism; mass spectrometry

## 1. Introduction

Propionic acidemia (PA) is a rare, autosomal recessive disorder engendered by a dysfunctional propionyl-coA carboxylase (PCC) enzyme [1]. The impairment of PCC is caused by mutations in *PCCA* or *PCCB*, genes which encode for the alpha and beta subunits of the 750 kDa heterododecamer [2,3]. PA has an estimated incidence of 1:100,000 live births in the general population, and a higher prevalence in isolated populations like the Amish Mennonite community due to the founder effect [3,4]. Untreated, PA can lead to an accumulation of propionyl-CoA metabolites, causing hypotonia, seizures, dehydration, and in severe cases, coma and death.

PCC catalyzes the carboxylation of propionyl-CoA to form methylmalonyl-CoA, which in turn can be converted to succinyl-CoA and utilized in the tricarboxylic acid cycle (TCA) [5]. This reaction represents a crucial step in the catalysis of isoleucine, valine, threonine and methionine for energy production. In PA patients, PCC dysfunction inhibits the conversion of propionyl-CoA to methylmalonyl-CoA, causing the accumulation and alternative shunting of propionyl-CoA into propionic acid. In addition to impacting energy metabolism by impairing the flow of amino acid catabolites into the TCA, the buildup of propionyl-CoA metabolites in PA has been shown to alter the activity of various

metabolic processes [2,6]. Notably, propionyl-CoA buildup can impact key enzymes of energy metabolism, including succinate dehydrogenase (SDHG), a-ketoglutarate dehydrogenase (aKDHG), pyruvate dehydrogenase (PDH), and succinyl-CoA ligase [7–9]. Various studies have asserted that the impairment of essential TCA enzymes leads to secondary mitochondrial dysfunction and increased oxidative stress in individuals with PA [10,11].

Although the current understanding of PA has deepened over past decades, leading to improved management of the condition, major complications still arise in many cases of PA [12]. The state of preventative and interventional treatments for PA remains inadequate in mitigating these complications. Thus, fully understanding the effects of PA on system-wide processes is essential in developing new interventions for the disease.

Very few comprehensive metabolomics studies have been published on samples from patients with this condition, exclusively testing plasma, urine, whole blood from dried blood spots or fibroblasts [13–15]. Moreover, to our knowledge there have been no metabolomic studies on the red blood cells (RBCs) of individuals afflicted by this disease. RBCs are the most abundant cell type in the body, and are critical in gas exchange and systemic homeostasis [16,17]. Metabolic analysis of blood is at the core of clinical chemistry, routinely informing medical efforts to diagnose and treat disease [18]. Because pancytopenia and anemia are associated with PA, investigating the disease-specific impacts on RBC metabolism is essential for fully understanding the pathophysiology of this disorder [2,19]. In order to characterize the metabolic effects of PA on red cells, we performed a comprehensive metabolomic analysis on RBC samples from PA patients.

## 2. Materials and Methods

### 2.1. Sample Collection and Processing

Samples were collected through venipuncture from 10 healthy controls and 3 patients with PA at the Central Pennsylvania Clinic under institutionally reviewed Protocol No. 2014-12 and upon signing of informed consent. The three PA patients included a 24-yo who has had 3 episodes of life-threatening cardiomyopathy and two children, 4 and 6 years of age, who have mildly depressed left ventricular ejection fractions 50–55% (nl 65–70%) and both have increased end diastolic volumes and mildly increased beta-naturetic peptides. RBCs were separated from whole blood through centrifugation for 10 min at 4 °C and 2000× *g*.

### 2.2. Mass Spectrometry-Based Metabolomics

RBCs were extracted for metabolomic analysis as previously described [20–22]. Samples were resuspended at $-4$ °C in ice-cold lysis buffer (5:3:2 MeOH: ACN: $H_2O$) at a 1:10 RBC:buffer ratio. Next, the samples were vortexed vigorously for 30 min at 4 °C. Solids were separated from the extract with centrifugation for 10 min at 18,213 rcf and 4 °C and discarded. The supernatant was analyzed with ultra-high-pressure liquid chromatography coupled to mass spectrometry at a flow rate of 450 μL/min using 5 min gradients as previously described (UHPLC-MS—Vanquish and Q Exactive, Thermo Fisher, San Jose, CA, USA) [23]. After untargeted acquisition, metabolite peaks were manually validated using the software Maven (Princeton University, Princeton, NJ, USA) and the KEGG database. Peak quality was determined by assessing $^{13}C$ abundance, blanks, and technical mixes. Statistical analyses, including principal component analysis (PCA), hierarchical clustering analysis, *t*-test and determination of variable importance in projection (VIP) for metabolites with the highest loading weights on PC1 were performed through MetaboAnalyst 5.0, as in previous studies [22–25].

## 3. Results

### 3.1. PA RBCs Show A Unique Metabolic Signature When Compared with Controls

Metabolomics analyses were performed on RBC samples from PA patients and compared with unaffected controls (Figure 1A; Supplementary Table S1). Distinct metabolic phenotypes between the two groups were confirmed by principal component analysis

(PCA—Figure 1B), which discriminated the two groups across PC1, which explained 32.6% of the total variance. A variable importance in projection analysis (VIP) and hierarchal clustering analysis (HCA) were performed on the data to identify the metabolic pathways most affected in RBCs from PA patients compared to controls (Figure 1C,D). The VIP analysis revealed the top 15 most influential metabolites on the clustering pattern, highlighting compounds from late glycolysis, the pentose phosphate pathway (PPP), purine and amino acid metabolism (Figure 1C). The HCA provided an expanded view of the metabolic differences between PA and control samples, displaying trends of the top 50 significant metabolites by ANOVA and highlighting additional differences in fatty acid and mitochondrial metabolism (Figure 1D).

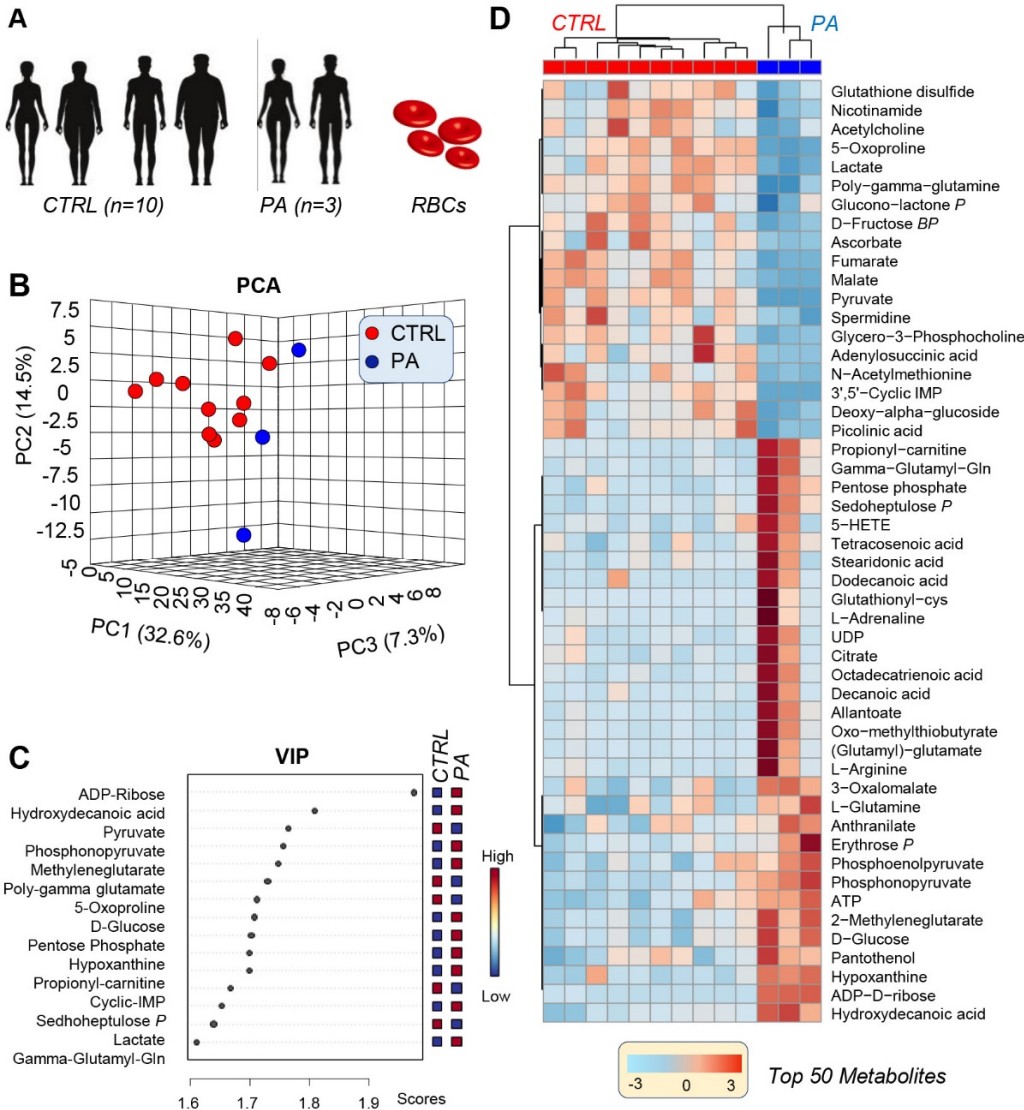

**Figure 1. Multivariate analyses show a distinct metabolic signature of PA RBCs.** (**A**) Untargeted metabolomics analyses were carried out on RBC samples from 3 PA patients and 10 unaffected CTL patients using UHPLC-MS. (**B**) A principal component analysis (PCA) indicated that PA samples were metabolically distinct when compared to CTL samples. (**C**) The variable importance in projection (VIP) values from a partial least square discriminant analysis (PLS-DA) highlighted the metabolites that contributed most to the clustering pattern, notably metabolites of glucose metabolism, fatty acid metabolism, and purine metabolism. (**D**) A hierarchal clustering analysis (HCA) of the top 50 significant metabolites by ANOVA revealed a broader picture of the metabolic alterations between PA and CTL samples.

### 3.2. PA Samples Show Alterations of Glycolysis and the PPP

The PA samples showed significant differences in glycolysis (Figure 2), especially in the later steps of the pathway. Though intracellular glucose was significantly elevated in the PA RBC, fructose-1, 6-bisphosphate was significantly lowered (Figure 2A). Phosphoenolpyruvate was increased in the PA samples, but its downstream products, pyruvate and lactate, were significantly decreased (Figure 2A). Phosphonopyruvate, however, was significantly increased, perhaps suggesting diverted flux into phosphonate metabolism (Figure 2A), or altered pyruvate kinase activity. Despite apparent lower levels of glycolytic intermediate and byproducts, ATP levels were found to be higher in PA subjects in this cohort (Figure 2A).

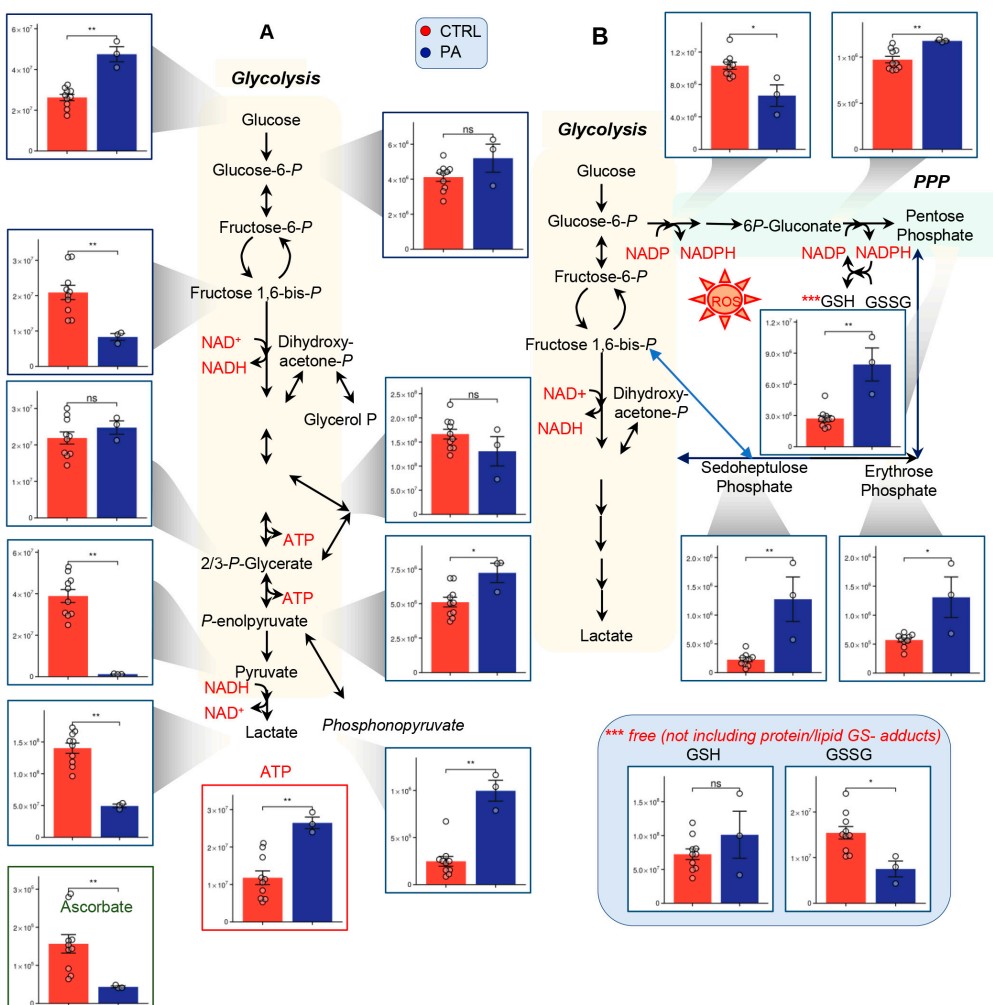

**Figure 2. PA remodels (A) glycolysis and the (B) pentose phosphate pathway in RBCs.** Y-axes represent peak area (arbitrary units). Asterisks indicate significant results, error bars represent ± standard error of the mean (unpaired *t*-test, 2-tailed distribution, * $p < 0.05$, ** $p < 0.01$).

With the exception of 6-phosphogluconolactone, the metabolomic analyses of the PA samples revealed higher levels of most PPP metabolites, when compared to the controls (e.g., 6-phosphogluconate and ribose phosphate/pentose phosphate isobars and non-oxidative phase erythrose phosphate and sedoheptulose phosphate—Figure 2B). Apparent activation of the PPP, as suggested by steady-state data, corresponded to decreases in GSSG (oxidized glutathione) levels, suggestive of potential increases in PPP-derived NADPH-dependent reduction. (Figure 2B). However, significant decreases in ascorbate were noted in PA patients (Figure 2A). Combined with the decreases in glycolysis and activation of the PPP, these data are, overall, suggestive of a higher basal oxidant stress level in

these RBCs which is counteracted by endogenous antioxidant systems in the absence of additional stressors.

### 3.3. Alterations of the TCA Suggest Mitochondrial Disorder in PA Patients

While devoid of mitochondria, mature RBCs harbor several cytosolic isoforms of Krebs cycle enzymes that can metabolize carboxylic acids and participate in redox homeostasis via the reduction/oxidation of key cofactors (e.g., NADH, NADPH) [24,26]. While the accumulation of propionate is a hallmark of PA, little is known about carboxylate metabolism in the mature RBC from PA patients. In addition, altered levels of mitochondria-containing reticulocytes in the bloodstream of PA patients may impact carboxylate metabolism in blood [19]. Consistent with a nonfunctional propionyl-CoA carboxylase (PCC), increases in carnitine-conjugated propionate were observed in PA patients (Figure 3A), along with decreases in branched chain amino acid (BCAA) precursors to propionyl-CoA (Figure 3A). In keeping with decreased propionyl-CoA catabolism, succinate appeared to be slightly decreased in PA patients (Figure 3). Moreover, other carboxylates (fumarate and malate) that can be generated in RBCs by alternative pyruvate catabolism and purine salvage reactions, were significantly decreased in the PA samples when compared with controls (Figure 3) [26,27]. Glutamine was increased in the PA samples, with no evident decreases in glutamate (Figure 3).

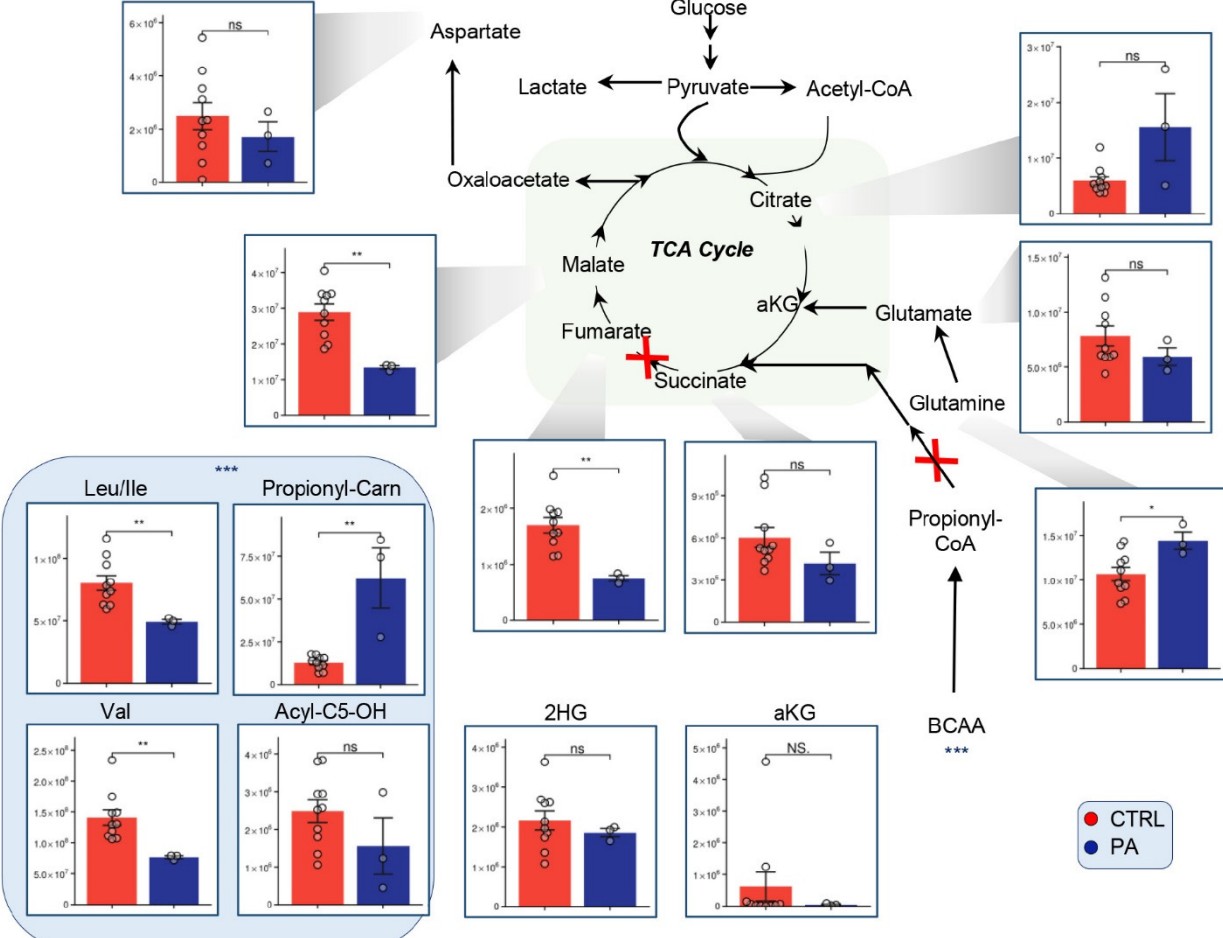

**Figure 3. PA alters levels of TCA metabolites and precursors in RBCs.** Summary of the tricarboxylic acid cycle (TCA) and related metabolites in PA patients relative to controls. Y-axes represent peak area (arbitrary units). Asterisks indicate significant results, error bars represent ± standard error of the mean (unpaired *t*-test, 2-tailed distribution, * $p < 0.05$, ** $p < 0.01$, *** $p < 0.001$).

### 3.4. Methionine, Arginine, and Purine Metabolism Are Altered in PA Samples Relative to Controls

In light of the observed changes in ATP, carboxylic acids and BCAAs, we then focused on the pathways that cross-regulate the homeostasis of the metabolites above, including methionine, arginine and purine metabolism (Figure 4A).

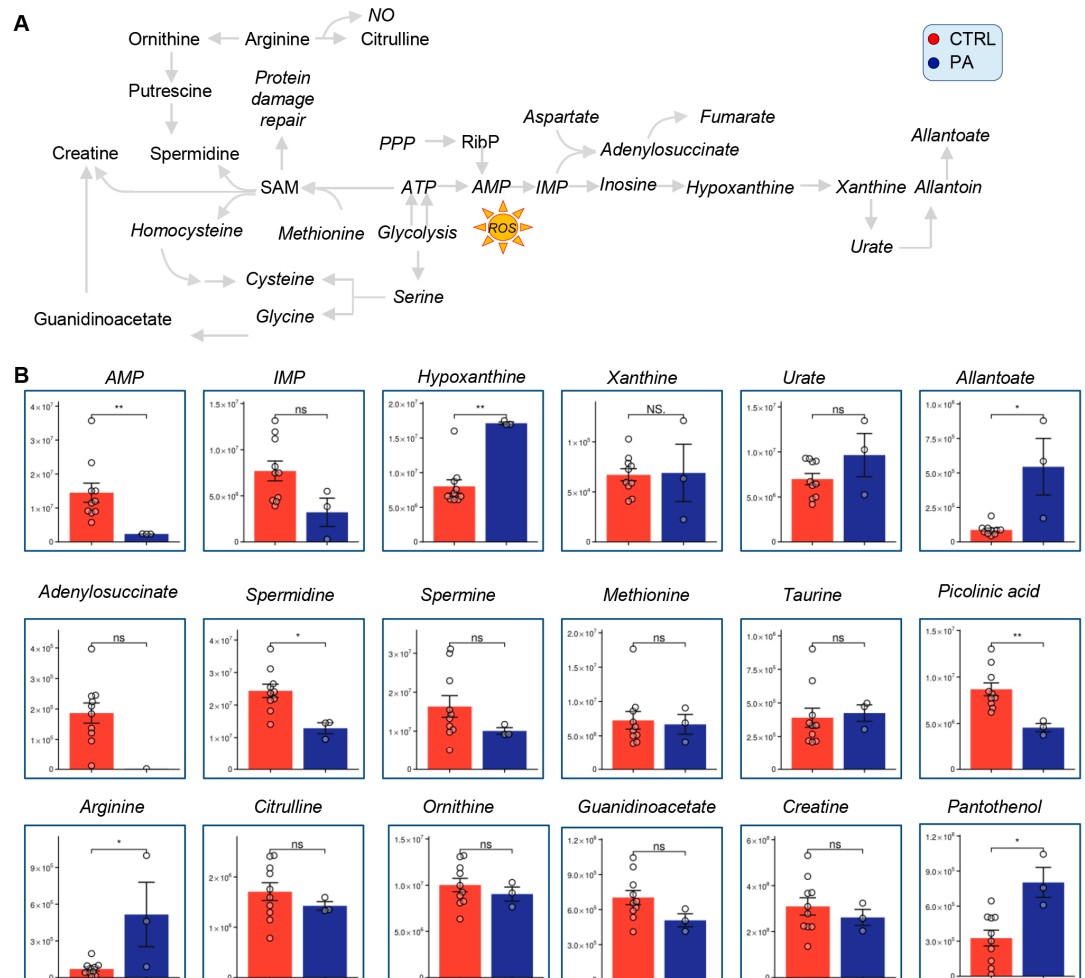

**Figure 4. PA RBCs show different levels of purine and arginine metabolism indicating altered oxidative stress.** Summary of purine, arginine, and methionine metabolism (**A**) and bar plots with superimposed dot plots of key metabo-lites in these pathways (**B**). Y-axes represent peak area (arbitrary units). Asterisks indicate significant results, error bars represent ± standard error of the mean (unpaired *t*-test, 2-tailed distribution, * $p < 0.05$, ** $p < 0.01$).

With respect to purine metabolism, the PA samples showed significantly decreased AMP, and increases in its downstream oxidation products, hypoxanthine and allantoate (Figure 4B). While not reaching significance, it is interesting to note that the purine salvage intermediate adenylosuccinate was only detected in control RBCs, but not in erythrocytes from PA patients (Figure 4B), in keeping with decreases in fumarate levels—suggestive of downregulated purine salvage in PA RBCs. Additionally, spermidine was significantly decreased in the PA samples. Arginine was significantly increased in the PA samples, though its downstream products of catabolism ornithine and citrulline were unchanged. Picolinic acid, a neurotoxic tryptophan oxidation product, was significantly decreased in the PA samples (Figure 4B). However, no significant changes were observed in the markers of redox homeostasis/oxidant stress-induced protein damage repair, methionine and taurine [28,29]. Finally, pantothenol, a CoA precursor, was increased in the PA samples relative to controls.

### 3.5. Fatty Acid Metabolism Is Altered in PA RBCs

Metabolomics analyses revealed lower levels of several markers of lipid peroxidation in the PA samples relative to controls, including arachidonic and linoleic acid oxidation products—HETEs and HoDEs (Figure 5). The levels of long chain fatty acids (LCFA) appeared comparable between PA and control samples with the exception of GLA, which was significantly elevated in PA red cells. Short chain fatty acids (SCFA) and medium chain fatty acids (MCFA) were more abundant in the PA samples than control. Lastly, glycerol phosphate, a precursor to glycerolipid biosynthesis, was decreased in PA patients.

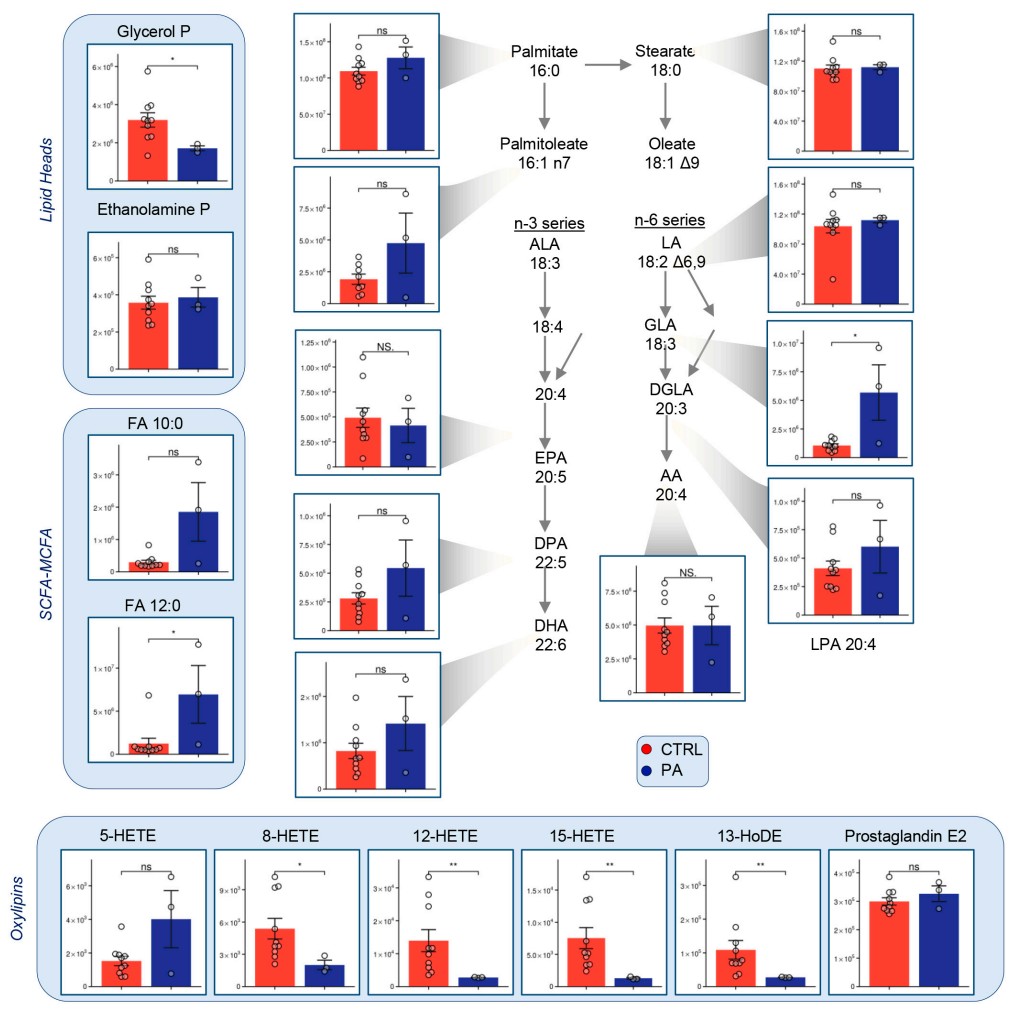

**Figure 5. PA remodels fatty acid metabolism in RBC.** Y-axes represent peak area (arbitrary units). Asterisks indicate significant results, error bars represent $\pm$ standard error of the mean (unpaired *t*-test, 2-tailed distribution, * $p < 0.05$, ** $p < 0.01$).

### 3.6. PA RBCs Show Minimal Alterations in Markers of Hypoxic Metabolic Reprogramming

Though there were few significant changes in metabolites indicative of systems-wide hypoxic response, malate was significantly decreased in PA RBCs relative to control (Figure 6). Glutamate, a-ketoglutarate (aKG), 2-hydroxyglutarate (2HG), creatinine, and sphingosine-1-phosphate (S1P) were slightly decreased in the PA samples, but these changes were not statistically significant.

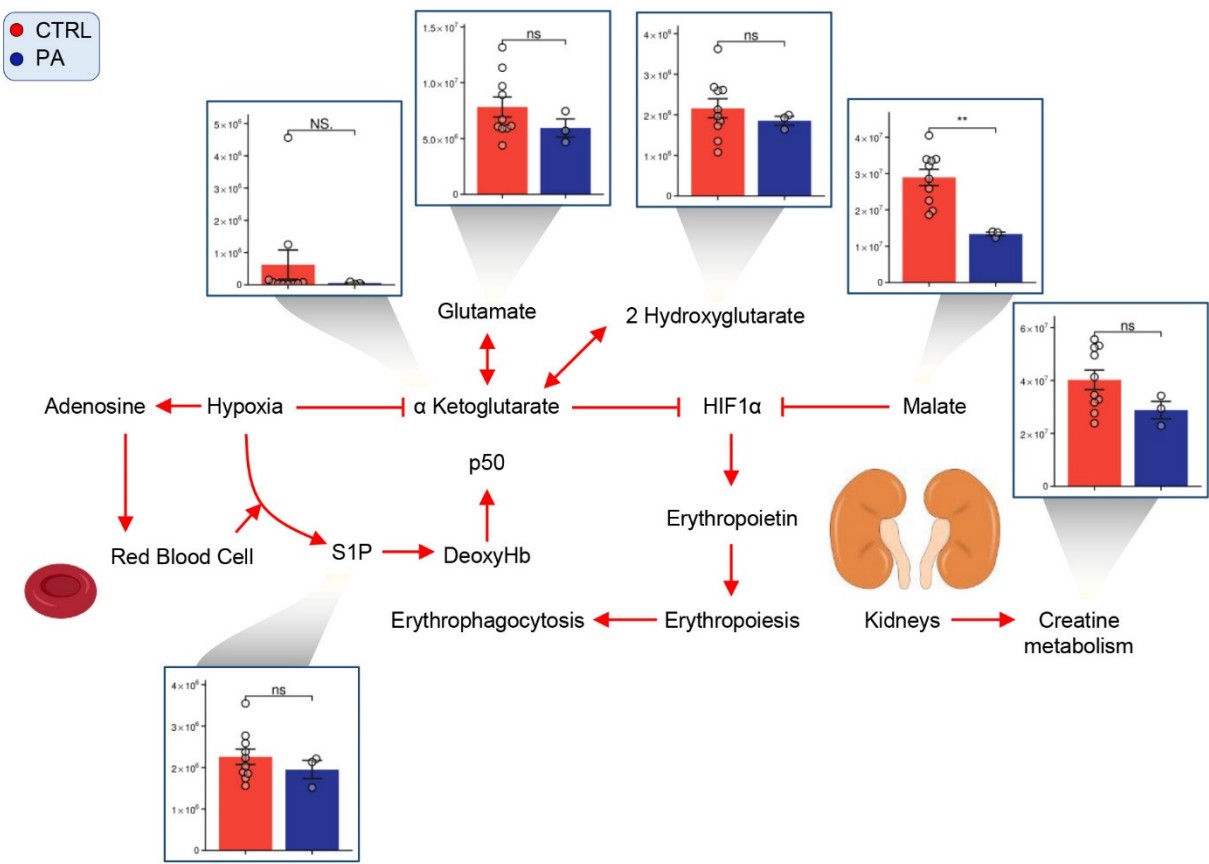

**Figure 6. PA RBCs do not show a distinct hypoxic response.** Summary of hypoxia-induced signaling pathways in PA red cells. Y-axes represent peak area (arbitrary units). Asterisks indicate significant results, error bars represent ± standard error of the mean (unpaired *t*-test, 2-tailed distribution, ** $p < 0.01$).

## 4. Discussion

In the present study we described the metabolic impact of PA on RBCs. The investigation of RBCs in the context of PA is relevant in that the condition is associated with anemia and impaired hematopoiesis [19]. Given the absence of mitochondria in mature RBCs, most studies have focused on the impact of PA on hematopoietic precursors, in which mitochondrial metabolism is an essential contributor to proliferation and differentiation [30]. During erythropoiesis, propionyl-CoA catabolites, such as succinyl-CoA, participate in heme synthesis, though the dominant succinyl-CoA metabolic route is via glutaminolysis [31,32]. Despite an important body of literature on this topic, little is known about mature RBC metabolism in the context of PA. Here we report that PA is associated with a significant decrease in the steady-state level of glycolytic products, pyruvate and lactate. On the other hand, the apparent activation of the PPP was observed, with all the intermediates of both the oxidative and non-oxidative phase increasing in the PA group compared to controls. Previous studies have suggested that increased flux through the PPP in RBCs may constitute a compensatory response in the face of oxidative stress [33]. The PPP combats $O_2$ stress by generating reducing cofactors (NADPH) necessary to preserve the homeostasis of reduced glutathione (GSH) upon oxidant challenge, by recycling oxidized glutathione (GSSG) in the face of oxidant stress [33]. Organic acids like propionic acid have been shown to stymie mitochondrial GSH transport (perhaps in erythroid precursors or residual circulating reticulocytes), which may explain the reduction of GSH in PA patients [34]. Despite decreases in the steady state levels of glycolytic metabolites, we observed unexpected increases in ATP levels in the PA samples [35]. These results are consistent with the potential increase in reticulocytosis and residual mitochondrial metabolism in the mature RBCs from these subjects. More importantly, in the context of PA,

RBC levels of carboxylic acids are a good indicator of systemic responses to PCC deficiency, as well as to mitochondria-targeting stressors, such as physiologic (e.g., high altitude) or pathological hypoxia (e.g., ischemia or chronic kidney disease) [36–38]. As such, we next focused on carboxylic acid metabolism in both control and PA subjects.

Because PCC is a key player in branched-chain amino acid (BCAA) degradation, we expected that its dysfunction would result in an accumulation of upstream products including BCAA. Interestingly, the opposite was true; leucine/isoleucine and valine had lower levels in the PA samples relative to controls. Because BCAAs generate high levels of propionic acid in PA patients, a diet restricted in these amino acids is often used to treat the condition [39]. As such, these reduced levels may reflect lower dietary intake of BCAAs in the PA patients. Corroborating previous studies, C3 carnitine, a direct metabolite of propionic acid, was increased in the PA samples [14]. Decreases in the RBC levels of succinate may reflect impaired PCC and the reduced entry of succinyl-CoA into the TCA. Alternatively, this observation may also indicate diminished succinate dehydrogenase activity (SDH) [14] in mitochondria-containing cells that is reflected in the RBC metabolism because of alterations in the circulating levels of these metabolites. Altered circulating levels of succinate are then deemed to be reflected in RBCs, as well as, for example, in response to hemorrhagic hypoxia [37]. Although mature erythrocytes do not contain mitochondria and other organelles, cytosolic isoforms of TCA enzymes and metabolites have been shown to be present in red cells, where they participate in the homeostasis of reducing equivalents [26]. This data may also reflect TCA activity in mitochondria-containing reticulocytes, which would contribute to explaining the increases in the levels of dicarboxylates downstream to succinate in the TCA cycle, such as fumarate and malate.

Alterations in the levels of fumarate have been previously linked to altered purine salvage following purine deamination in response to hypoxia/oxidant stress [27]. Of note, previous studies in RBC storage have correlated decreases in hypoxanthine and allantoate, and increases in spermidine with increased stress, the opposite of what was observed in the PA samples [27,40] and decreased capacity of the RBC to circulate in vivo [27,41].

The alteration of short chain fatty acids (free and carnitine-conjugated propionate) was here accompanied by the accumulation of some medium chain and long chain fatty acids (dodecanoic and γ-linolenic acid), as well as a decrease in lipid peroxidation (markers of RBC splenic sequestration and in vivo survival) [42]. These results contrast with previous research that has shown an increase in odd-numbered LCFA, but with the exception of GLA, support studies which assert no change in long-chain poly unsaturated fatty acids (LCPUFA) in the PA erythrocytes [14,43,44]. On the other hand, our results are interesting in that they follow opposite trends to those reported recently in the context of storage and the oxidant stress-induced activation of fatty acid desaturases, suggesting that in the absence of additional stressors, RBCs from PA patients are capable of withstanding oxidant stress through the basal activation of other redox pathways (e.g., the PPP) [45]. Since the rate-limiting enzyme of the PPP is glucose 6-phosphate dehydrogenase, which is coded by a gene on chromosome X, it is interesting to speculate whether G6PD dosage might impact the severity of PA in a sex-dependent fashion, with RBCs from females predicted to be better protected from oxidant stress than males carrying the PCC deficiency [22,46]. It would be interesting to investigate whether dietary supplements aimed at leveraging the apparent overactivation of both the oxidative and nonoxidative phase of the PPP (e.g., inosine from rejuvenation solutions) could be used to further contribute to redox and energy homeostasis in individuals with PA.

This study holds several limitations, including the exploratory nature of the analysis (semitargeted approach) on a limited size cohort, which limits the opportunity to gain readily translatable, clinical insights to tackle the impact of PA on this patient population. Future studies will expand upon the preliminary analyses presented here, not only by enrolling additional subjects to prospectively validate the molecular signatures identified here and link molecular signatures to clinical phenotypes and interventions, but also by including stable isotope labeled tracers (e.g., 1,2,3-$^{13}C_3$-glucose; $^{13}C$-BCAA) to complement

the steady-state observations described herein with data on metabolic fluxes and the alterations thereof, in the context of PA [47,48]. Despite the exploratory nature of this study on a small cohort, using advanced omics technologies to shed light on metabolic abnormalities targeting underrepresented minorities in academic research (the Amish Mennonite community), paving the way for follow-up direct targeting of critical pathways found herein to be dysregulated in the context of PA (glycolysis, PPP, purine oxidation and carboxylate metabolism).

**Supplementary Materials:** The following are available online at https://www.mdpi.com/article/10.3390/separations8090142/s1, Table S1: PA—raw metabolomics data.

**Author Contributions:** Conceptualization, H.M. and A.D.; methodology, M.K.R., F.I.C., F.G., A.D.; formal analysis, M.K.R., A.D.; resources, G.O., H.M., A.D.; data curation, F.I.C., F.G.; writing—original draft preparation, M.K.R., A.D.; writing—review and editing, all authors.; visualization, A.D.; supervision, H.M., A.D.; project administration, H.M., A.D.; funding acquisition, A.D. All authors have read and agreed to the published version of the manuscript.

**Funding:** This research was supported by funds from the RM1GM131968 (ADA) from the National Institute of General and Medical Sciences, and R01HL146442 (ADA), R01HL149714 (ADA), R01HL148151 (ADA), R21HL150032 (ADA), from the National Heart, Lung, and Blood Institute.

**Institutional Review Board Statement:** Samples were collected at the Central Pennsylvania Clinic under institutionally reviewed Protocol No. 2014-12 and upon signing of informed consent.

**Informed Consent Statement:** Informed consent was obtained from all subjects involved in the study.

**Data Availability Statement:** All raw data are provided in Supplementary Table S1.

**Conflicts of Interest:** Though unrelated to the contents of this manuscript, the authors declare that AD is founder of Omix Technologies Inc and Altis Biosciences LLC. He is also an advisory board member for Hemanext Inc and Forma Therapeutics Inc, and a consultant for Rubius Therapeutics.

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
