# Peer review of "Red Blood Cell Metabolism in Patients with Propionic Acidemia"

_separations, doi:10.3390/separations8090142_

Round 1

Reviewer 1 Report

Reviewer 1

Journal: Separations (MDPI)

Manuscript ID: Separations-1347906

Title: Red blood cell metabolism in patients with propionic acidemia

Authors: Micaela K Roy, Francesca Isabelle Cendali , Gabrielle Ooyama , Fabia Gamboni , Holmes Morton, Angelo D'Alessandro 

Comments to the Authors:

Summary

Propionic acidemia is a rare inherited autosomal recessive disorder characterized by the accumulation of propionic acid due to propionyl-COA carboxylase deficiency. This severe condition is manifested clinically at all ages including in neonates by metabolic acidosis and hyperammonemia that can be life-threatening or lead to neurological damage and cardiomyopathic complications in adult life. Despite the existence of effective therapy via a low protein diet and carnitine, the overall outcome of PA remains poor.  Clearly, more research is required to address the gaps in the literature which will enable effective, personalized, and targeted therapies in these patients and ultimately improve the survival rate and quality of life. Given that very few metabolomic studies have been undertaken in PA, Roy et al, conducted a study investigating the metabolic effect of this condition in erythrocytes of PA patients. According to the authors, putative metabolites recovered were associated with abnormalities in the TCA cycle, pentose phosphate pathway, lipid, purine, and BCAA metabolism as well as increased oxidative stress.

This study is important in providing new evidence to the literature database that will aid clinicians in the identification of PA in patients, early prognosis along with efficient treatment and management of long-term symptoms.

Comments:

A well-written manuscript, with clear comprehensible diagrams presenting potential metabolic pathways involved in PA. The authors are to be commended. However, there are a few issues that require amendment which will improve understanding of the study design and transparency. One must keep in mind that not all readers are experts in the field of metabolomics or PA.

Title: Given that this study is one of the first blood metabolomic studies to identify a distinct metabolic signature in PA patients, I would suggest that the title is changed to signify this.

Perhaps in the line of “Blood metabolomic profiling reveals predictive signatures associated with PA.” or “Untargeted metabolomics identifies a metabolic fingerprint for PA”

Abstract

Define RBC, PPP, PKD, and TCA.

Main Manuscript

Introduction

Line 58 ‘Very few comprehensive metabolomics studies have been published on samples from patients with this condition” Add references to existing studies

Materials and Methods

Line 71, more details are required to indicate the study design, describe the population under investigation e.g children, adults, or neonate mean age, sex, ethnic background. Did this sample include members of the Amish-Mennonite community? It is not clear from the Methods section.

Discussion ‘

Line 240 add reference

Limitations

-Provide more details on the limitations of your study. For example the advantages /disadvantages of the metabolomic technique that was applied (e.g untargeted).

Author Response

Responses to Reviewer 1

Comments: A well-written manuscript, with clear comprehensible diagrams presenting potential metabolic pathways involved in PA. The authors are to be commended. However, there are a few issues that require amendment which will improve understanding of the study design and transparency. One must keep in mind that not all readers are experts in the field of metabolomics or PA.

Authors’ reply: Thank you for the kind and constructive comments on the manuscript!

Title: Given that this study is one of the first blood metabolomic studies to identify a distinct metabolic signature in PA patients, I would suggest that the title is changed to signify this.

Perhaps in the line of “Blood metabolomic profiling reveals predictive signatures associated with PA.” or “Untargeted metabolomics identifies a metabolic fingerprint for PA”

Authors’ reply: Thank you for the suggestion. In this study, we did not perform a formal biomarker study, since we had no validation cohort we could access at that stage. While working to bridge this gap, we would like to avoid adopting the Reviewer’s kind suggestion, to prevent overstating the relevance of our exploratory work. Thank you in any case for the generous suggestion!

Abstract

Define RBC, PPP, PKD, and TCA.

Authors’ reply: Now defined at first appearance.

Main Manuscript

Introduction Line 58 ‘Very few comprehensive metabolomics studies have been published on samples from patients with this condition” Add references to existing studies

Authors’ reply: Now we added the following references:

https://pubmed.ncbi.nlm.nih.gov/32143654/ 

Materials and Methods

Line 71, more details are required to indicate the study design, describe the population under investigation e.g children, adults, or neonate mean age, sex, ethnic background. Did this sample include members of the Amish-Mennonite community? It is not clear from the Methods section.

Authors’ reply: We now expanded the description of the population, within the limits imposed by our IRB.

Discussion ‘ Line 240 add reference

Authors’ reply: Reference added

Limitations -Provide more details on the limitations of your study. For example the advantages /disadvantages of the metabolomic technique that was applied (e.g untargeted).

Authors’ reply: Limitations have been added.

Reviewer 2 Report

This is a very concise, nicely written manuscript. The authors compared the metabolism between RBCs from healthy subjects and subjects with propionic acidemia. Although this is a purely observational study, and some of the findings are kind of expected, still it does not hurt its impact in the field, since no literature documented the red cell abnormalities. Therefore, I suggest the publication following a minor revision.

  1. There are some typos, miss of spacing, and formatting issues. I suggest multiple authors go through the manuscript to fix those issues. If an abbreviation is used, spell out in its first appear.
  2. The authors need to improve the methods section. There are some analysis methods missing. It is understood how the experiments were performed, but how about the statistics? And I am not sure how Fig 1. B and C were generated. How was the PCA carried out, and what does the score in Fig. 1C mean?
  3. While it is plausible that the authors tried hard to explain why they were observing those results in the discussion section, it would be more valuable to discuss how some of the findings could help uncover mechanisms associated with critical health issues in order to shed new insights on therapeutic interventions.
  4. Just out of curiosity, do the authors think those metabolic abnormalities in red cells would affect their biorheological functions? And further link to some microcirculatory problems and symptoms?
  5. The authors did mention some of the observations were attributed to reticulocytes instead of mature red cells. If the authors were to compare reticulocyte-poor red cell samples between healthy and diseased people, how would it affect the conclusions of this study?

Author Response

Responses to Reviewer 2

This is a very concise, nicely written manuscript. The authors compared the metabolism between RBCs from healthy subjects and subjects with propionic acidemia. Although this is a purely observational study, and some of the findings are kind of expected, still it does not hurt its impact in the field, since no literature documented the red cell abnormalities. Therefore, I suggest the publication following a minor revision.

Authors’ reply: Thank you for the kind and constructive comments on the manuscript!

Reviewer’s comment: There are some typos, miss of spacing, and formatting issues. I suggest multiple authors go through the manuscript to fix those issues. If an abbreviation is used, spell out in its first appear.

Authors’ reply: The paper has been revised to address these minor concerns. Apologies for these issues!

Reviewer’s comment: The authors need to improve the methods section. There are some analysis methods missing. It is understood how the experiments were performed, but how about the statistics? And I am not sure how Fig 1. B and C were generated. How was the PCA carried out, and what does the score in Fig. 1C mean?

Authors’ reply: We now provide a short explanation of the software used for the standard metabolomics analyses we performed here. The workflow is pretty much identical to that used by most authors in the field, including our previous metabolomics studies on RBCs (e.g., D’Alessandro et al. Haematologica 2021; Stefanoni et al. Haematologica 2020; etc etc).

Reviewer’s comment:  While it is plausible that the authors tried hard to explain why they were observing those results in the discussion section, it would be more valuable to discuss how some of the findings could help uncover mechanisms associated with critical health issues in order to shed new insights on therapeutic interventions.

Authors’ reply: This is a point we wholeheartedly agree with. Unfortunately, we were afraid to extrapolate such translational relevance from an exploratory study on such a small cohort. We now refer to this aspect of the study as a limitation that we plan on addressing in our upcoming studies (currently enrolling).

Reviewer’s comment: Just out of curiosity, do the authors think those metabolic abnormalities in red cells would affect their biorheological functions? And further link to some microcirculatory problems and symptoms?

Authors’ reply: We can only speculate at this stage. However, this is a great point that we will address in future studies in which we will also try to perform measurements of RBC function and morphology.